# Predictors of internalised stigma among people with mental illness attending a psychiatry outpatient clinic in Ethiopia: Institution based cross sectional study

Wondale Getinet Alemu[1,2]*, Lillian Mwanri[3], Clemence Due[4], Telake Azale[5], Anna Ziersch[1]

**1** College of Medicine and Public Health, Flinders Health and Medical Research Institute, Flinders University Adelaide, Australia, **2** Department of Psychiatry, College of Medicine and Health Sciences, University of Gondar, Gondar, Ethiopia, **3** Research Centre for Public Health, Equity, and Human Flourishing, Torrens University Australia, Adelaide Campus, South Australia, **4** School of Psychology, The University of Adelaide, Adelaide, Australia, **5** Institute of Public Health, College of Medicine and Health Sciences, University of Gondar, Gondar, Ethiopia

\* geti0001@flinders.edu.au

## Abstract

### Background

Despite initiatives to increase access to mental health care and improve the quality of life for individuals living with mental illness, there is limited information on internalized stigma and its impact on these individuals. This study aimed to determine the prevalence of internalised stigma and identify associated factors (sociodemographic, clinical, and substance use) among people with mental illness attending an outpatient clinic in Ethiopia.

### Method

Institution-based cross-sectional study was conducted with patients with mental illness at the University of Gondar Hospital clinic. We recruited 638 participants from the clinic using systematic random sampling with an interval of three applied. Internalised stigma was measured using the nine-item (ISMI-9) Internalised stigma of Mental Illness Scale. Variables were coded and entered into SPSS-28 software for further analysis. To analyze the data, we used descriptive and multivariate logistic regression analysis. Adjusted odds ratio (AOR) with 95% confidence interval (CI) and p-value less than 0.05 were considered significant.

### Results

Prevalence of internalised stigma among study participants was 49.1% (95% CI: 45, 52). The following attributes were associated with a greater likelihood of high internalised stigma, participants with no formal education (AOR=2.19, 95% CI:1.33, 3.61); patients with fair self-reported health (AOR=3.12, 95% CI:1.28, 7.59), patients with poor self-reported health (AOR= 9.11, 95% CI: 2.89, 28.73), patients with suicidal ideation

**Data availability statement:** All relevant data are within the paper and its Supporting Information files.

**Funding:** The author(s) received no specific funding for this work.

**Competing interests:** The authors have no competing interest.

**Abbreviations** AOR: adjusted odd ratio; CI: confidence interval; COR: crude odd ratio; CGI: Clinical Global Impression; ISMI: Internalised stigma of mental illness; LMIC: Low and middle-income countries; MARS: Medication adherence scale; OR: Odd ratio; QoL: Quality of life; SD: Standard deviation; SPSS: Statistical package for social sciences; WHO: World health organization.

(AOR=1.95, 95% CI:1.37, 2.79), alcohol users (AOR= 1.89, 95% CI:1.24,2.91), patient with low self-esteem (AOR=1.55, 95% CI:1.09, 2.21), patient with poor drug adherence (AOR=2.2, 95% CI:1.30,3.71), patients with family history of substance use (AOR= 2.46, 95% CI:1.54,3.93).

## Conclusions

The prevalence of high internalised stigma among patients with mental illness in was high. Therefore, anti-stigma activities, early outpatient support, drug adherence information, and reduction of suicidal behaviors are all necessary to reduce stigma in patients with mental illnesses.

## Background

Stigma relates to idea of discrediting, devaluation, and humiliation of an individual due to characteristics they possess [1,2]. Many aspects or forms of the stigma associated with mental health have been identified in the literature, but most common are internalised and perceived stigma [1]. People with mental illnesses endure shame, ostracism, and social exclusion, and this remains a global public health challenge [3,4] and significant number of people with severe mental illness face double challenges associated with symptoms and disabilities, and mental health related stigma and prejudices [5]. Patients with mental illness report that the effects of mental health stigma are worse than the symptoms of the disease itself [6]. Stigma associated with mental illness also makes access to mental health services more difficult [7]. Cultural values may influence how mental health related stigma operates and presents [8]. However, the impact of mental health-related stigma and discrimination are almost universal across countries [9]. For example, nearly 50% of people with schizophrenia who participated in a study involving twenty seven different countries said that they had experienced discrimination in their relationships and 67% of applicants for new jobs or persons looking for committed relationships reported being worried about discrimination [10].

As well as being carried by others in society, stigma can be internalised by the individual with the disease [11]. Internalised stigma is embracing and applying to oneself the unfavourable preconceptions that society associates with psychiatric illness [12–14]. Internalised stigma includes cognitive (self-defeating thoughts, sense of inferiority, sense of incompetence, negative self-perception), affective (feelings of despondency, sadness, embarrassment, shame, anger) and behavioural (self-stigmatization, self-isolation, concealment of status, social withdrawal, social avoidance) responses to perceived or experienced stigma [15–17].

Both perceived and internalised stigma may impact a person's disease and course of treatment, including their ability to receive suitable and competent medical care [18,19]. This includes that people who require care frequently do not seek services [20], people who start receiving care don't follow the suggested treatment plan [21], those with serious mental illnesses may not receive regular follow ups for their disease throughout the year [22]. In this way, stigma can isolate individuals and cause delays mental health care, which has a significant negative societal and economic impact [23]. The US surgeon General and WHO describe stigma as a significant barrier to successful engagement during treatment, including seeking, and maintaining services participation [24,25]. Stigma has also been reported to be the most challenging hurdle in the area of advancement on the field of mental illness and health in the future [26].

In terms of contributors to stigma, previous research suggests that stigma commonly results from a lack of knowledge, education, and understanding of the characteristics and difficulties of mental illness, such as unusual behaviors and aggression [27–29]. Other factors

that have been found to contribute to mental illness internalised stigma are younger age [30–32], male gender [33–40], unemployment [41–43], low social support [40–44], drug non-adherence [40,43], previous hospitalisations [41], low self-esteem [41,43], and having residual symptoms [43].

Ethiopia was selected as a study site as cultural perceptions and traditional beliefs about mental illness in Ethiopia often reinforce stigma and discrimination. Despite the high prevalence of mental health disorders in the country, mental health services remain underdeveloped and underutilized, making it essential to understand barriers such as stigma that hinder care-seeking and treatment adherence. Since most mental health services in Ethiopia are outpatient-based and concentrated in urban areas, outpatient clinics provide an ideal setting for studying people with mental illness who are actively receiving care but may still experience stigma. Furthermore, there is limited data on internalised stigma among Ethiopians with mental illness. Addressing this gap could provide valuable insights to inform programs aimed at reducing stigma and improving the quality of life for people with mental illness.

## Methods

### Study area

The study area is the Amhara region in Northwest Ethiopia, one of Ethiopia's regions with the highest population densities. The study was carried out at the University of Gondar Comprehensive Specialised Hospital in October to march, 2023. Gondar town had a total population of 395,000 in 2022 [45].

### Study design

An institution-based cross-sectional study was conducted. Data were collected through a face-to-face interview, including survey questions, reviewing the patient chart, and observing clinical symptoms.

### Study population

People with mental illnesses in the outpatient clinic at the University of Gondar Comprehensive Specialized Hospital participated. Patients treated for at least three months for any mental disorder were included. All the study participants are adult age above 18 years.

### Sample size determination

Single population proportion formula was employed using a 95% confidence level and 4% margin of error, a 10% nonresponse rate, and, considering a previous quality of life study in Ethiopia, 41% of people with mental illness [46]. Applying the formula: $n = (Z\alpha/2)^2 * P (1-P)/d^2$, where n is the minimum sample size required, Z is a standard normal distribution (Z=1.96) with a confidence interval of 95% and $\alpha = 0.05$. d is the absolute precision or tolerable margin of error (4%), P = estimated proportion is assumed as 41% (0.41)

Then n= $(1.96)^2 *(0.41) *(0.59)/ (0.04)^2$= 580, 10% non-response rates (580 *10/100) =58 Adding ten percent of nonresponse rate 580+58= 638

### Sampling technique

We used a systematic random sampling technique to obtain a total sample size of 638 patients who were followed up for the treatment of their mental illness from a group of 2400 patients who were followed, with a sample interval of three. Finally, 636 patients who had been followed up for at least three months and were 18 years old and older were included in the study.

Two patients did not complete the study after commencement. Patients who had a clinical diagnosis of schizophrenia, depression, bipolar disorder, anxiety, other psychotic disorders, stress and trauma-related disorders, or somatization disorders were eligible for inclusion.

## Data collection

The data was collected using a standardized questionnaire during a face-to-face interview at the outpatient psychiatry clinic. The questionnaire was prepared in English and translated into the local language, Amharic. Five psychiatry nurses and two MSc psychiatry supervisors participated in the data collection. The data collectors and supervisors received two days of training on data collection process, on the content of the questionnaire, interview methods, measurement techniques and participant approach. The data collectors and supervisor also participated in a practical session on demonstration of the interview and a pre-test before two weeks was conducted but these results were not included in the final analysis. Based on the findings from the pre-test, the questionnaires were revised and finalized. The interview was estimated to take 45 minutes. As part of the consent process, data collectors sought permission to access the person's health records, which could assist in providing background information of the patient's specific diagnosis, medications, and previous history of hospital admission. The data collectors are supervised daily by assigned supervisors, and the filled questionnaires are checked daily by the supervisors and principal investigator. Questionnaires were reviewed daily for completeness by data collectors, supervisors, and then by the researcher throughout data collection. Two incomplete surveys were discarded from the final analysis.

## Measurement

### Internalised stigma

The Internalised Stigma of Mental Illness (ISMI) scale was used to assess internalised stigma. The ISMI-9 contains nine items, which produce a total score. Items 2 and 9 are reverse coded before calculating the total score. The item scores are added and then divided by the total number of answered items. The resulting score should range from 1 to 4. For example, if someone answers eight of the nine items, the total score is produced by adding the eight responded items and dividing by 8. Finally, a mean score of 1.00-2.50 indicates the absence of high internalised stigma, and 2.51-4.00 indicates the presence of high internalised stigma [47]. Psychometric evaluation of the ISM-9 in the current sample showed high-scale reliability (Cronbach's alpha=0.88).

### Self-Esteem

A single item self-esteem scale included a one-item measure of global self-esteem used to evaluate an individual's self-esteem. This scale was created as a substitute for the Rosenberg Self-Esteem Scale. The single-item self-esteem scale is a measure of overall self-esteem. Participants rate the single item self-esteem (I have high self-esteem) on a 5-point Likert scale ranging from 1 (not very true of me) to 5 (very true of me). Despite being shortened, the scale has solid convergent validity with the Rosenberg Self-Esteem Scale and similar predictive validity [48].

### Medication adherence

The medication adherence scale (MARS-5) assesses patients standard treatment adherence through five questions and five level response formats (1=always, 2=often, 3=occasionally, 4=rarely, and 5=never). Responses were added for a total score ranging from 5 to 25, with

higher scores indicating greater adherence. We use the MARS-5 at a cutoff point greater than or equal to 20 as good adherence [49,50].

## Substance use

Patients who used certain substances like alcohol, khat, and cannabis (for non-medical purposes), in the last one year before data collection, were considered current substance users [51].

## The severity of illness

The severity of the disease was measured using the Clinical Global Impression (CGI) scale of subjective and objective measurement. The CGI scale has seven responses, with responses 1-3 indicating mild, four indicating moderate, and 5-7 indicating severe [52].

Other sociodemographic, clinical, and social support variables were measured with single item questions developed for the study.

## Data processing and analysis.

The data was entered into the SPSS-28 software for analysis. The outcome variable for analysis was internalised stigma. For several variables, descriptive statistics were employed. Using the mean internalised stigma score as the cutoff point, a binary logistic regression model was used to determine the association of factors with high internalized stigma. Variables with a p-value $\leq 0.2$ in the bivariate analysis was fitted into a multivariate logistic regression model to manage the impacts of confounding factors. Crude and adjusted odds ratio with 95% CI were calculated to determine the strength and presence of association. A p-value of $< 0.05$ was used to declare significance.

## Ethical consideration

The study adhered obtained ethical clearance from the Flinders University Human Research Ethics Committee with **Project No:** 5416 and the Institutional Review Board of the University of Gondar. Before their participation, written informed consent was diligently obtained from all study participants. Participants were fully briefed on the study's objectives and informed of their freedom to withdraw without any repercussions. Furthermore, to safeguard the confidentiality and privacy of the participants, the study employed code numbers instead of personal identification, ensuring that their personal information remained secure and undisclosed. These ethical measures underscore the commitment to the well-being and rights of the study of participants and are based on established ethical principles in research.

# Results

## Background characteristics

Of 638 invited participants for interviewer-administered questionnaires 636 (99.7%) completed the questionnaire, with the remaining two excluded from the final analysis due to incomplete responses. Around half of the respondents, 324 (50.9%) were female. On average, participants were 35.5 years old, with a SD of 11.7 years. Nearly half of participants, 274 (43.1%) had a diagnosis of schizophrenia. Figures for other diagnoses were depression 192 (30.2%), bipolar disorder 50 (7.9%), anxiety disorder 39 (6.1%), other psychotic disorders 68 (10.7%), stress/trauma-related disorder 06 (0.9%), somatization disorder 07 (1.1%) (Fig 1). Around half of 301 (47.3%) of the patients took only antipsychotic drugs, followed by 135 (21.2%) using both antipsychotics & antidepressants. Around 30% (199 (31.3%) were on

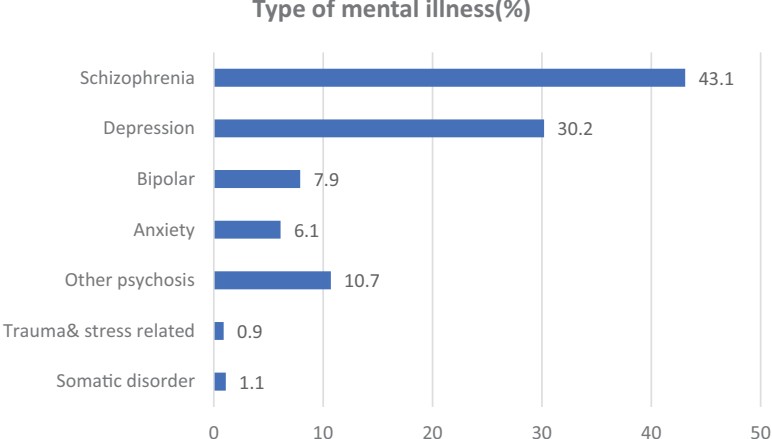

**Fig 1. Type of mental illness in psychiatry outpatient follow-up at University of Gondar Hospital, Ethiopia, 2023(n =636).**

follow up for more than five years, 316(49.7%) participants had low self-esteem, and almost one-tenth (13.7%) had poor drug adherence (Table 1, Fig 1).

## Prevalence of internalized stigma

The study showed a high level of internalised stigma in almost half of the participants – 312 (49.1%) (Fig 2). Among the participants, 44% strongly agreed that they can have a good fulfilling life despite their mental illness, 28.8% agreed that stereotypes about the mentally ill apply to them and 34% agree negative stereotypes about mental illness keep them isolated from the normal world, 33% agree they feel out of place in the world because they have a mental illness, 34% agree being around people who don't have a mental illness makes them feel out of place or inadequate, 34.4% agree that people without illness could not possibly understand them, 28.5% agree that nobody would be interested in getting close to them because they have a mental illness, 22% agree that they can't contribute anything to society because they have a mental illness and 44% of patients strongly agreed that they could have a good, fulfilling life, despite their mental illness (Table 2).

## Predictors of internalised stigma among people with mental illness in an outpatient clinic

In logistic regression analysis, sociodemographic factors (e.g. sex, religion, marital status, living condition, level of education, job of participant, residence, waiting time in clinics), clinical factors (mental illness, age of onset illness, duration of illness, number of episode/ yr., hospital admission, number of admission, comorbid illness, type of drug, drug side effect, counselling, duration of treatment, relapse, suicidal ideation, suicidal attempt, family history mental illness, family history substance use, family history suicide attempt, objective severity, subjective severity, personal perception of health), social support factors (r/ship with family, family participates in treatment, legal issues, self-esteem), and substance use factors(tobacco use, alcohol use, khat use, cannabis use, drug adherence) were examined.

Finally, in the multivariate logistic regression analysis, variables with a p value <0.2 were included in the initial model. Those found to be statistically significantly associated with high internalized stigma in the final model were no formal education, having suicidal ideation,

**Table 1. Sociodemographic, clinical, substance use, and social support related factors of people with mental illness attending Gondar Comprehensive Specialised Hospital outpatient clinic, 2023(n =636).**

| Variables | Categories | Frequency (n= 636) | Percept (%) |
|---|---|---|---|
| **Sociodemographic variables** | | | |
| Age | 18-34 | 326 | 51.3 |
| | ≥35 | 310 | 48.7 |
| Sex | Male | 317 | 49.8 |
| | Female | 319 | 50.2 |
| Religion | Orthodox | 497 | 78.1 |
| | Protestant | 21 | 3.3 |
| | Muslim | 115 | 18.1 |
| | Other | 03 | 0.5 |
| Marital status | Single | 249 | 39.2 |
| | Married | 253 | 39.8 |
| | Divorced | 114 | 17.9 |
| | Widowed/widower. | 20 | 3.1 |
| Living condition | Living alone | 95 | 14.9 |
| | Living with immediate family | 518 | 81.4 |
| | Living with other relatives | 10 | 1.6 |
| | Living in rehabilitation centers | 6 | 0.9 |
| | Other | 7 | 1.1 |
| Level of education | No formal education | 139 | 21.1 |
| | Reading & and writing but no formal education. | 46 | 7.2 |
| | Primary school (5-8) | 107 | 16.8 |
| | Secondary school (9-12) | 162 | 25.5 |
| | College/ university | 182 | 28.6 |
| Job of participant | Employed government. | 89 | 14 |
| | Private Employed | 52 | 8.2 |
| | Farmer | 101 | 15.9 |
| | Housewife | 104 | 16.4 |
| | Student | 72 | 11.3 |
| | Merchant | 81 | 12.7 |
| | No job | 127 | 20 |
| | Other | 10 | 1.6 |
| Residence | Rural | 205 | 32.2 |
| | Urban | 431 | 67.8 |
| waiting time in clinics | 30 minute-1hrs. | 534 | 84.0 |
| | 2hrs.-3hrs. | 102 | 16.0 |
| **Clinical variables** | | | |
| Mental illness | Schizophrenia | 274 | 43.1 |
| | Depressive disorder | 192 | 30.2 |
| | Bipolar disorder | 50 | 7.9 |
| | Anxiety disorder | 39 | 6.1 |
| | Other psychotic disorders | 68 | 10.7 |
| | Stress/trauma-related disorder | 6 | 0.9 |
| | Somatization disorder | 7 | 1.1 |
| Age of onset illness | </= 25yrs. | 287 | 45.1 |
| | >25yrs. | 349 | 54.9 |
| Duration of illness | 6 month-5yrs. | 401 | 63.1 |
| | 6yrs.-10yrs. | 141 | 22.2 |
| | >10yrs. | 94 | 14.8 |
| No, of episode/yr. | No episode | 298 | 46.9 |
| | 1episode | 212 | 33.3 |
| | >/=2episode | 126 | 19.8 |
| Hospital admission | Yes | 209 | 32.9 |
| | No | 427 | 67.1 |

*(Continued)*

**Table 1.** (Continued)

| Variables | Categories | Frequency (n= 636) | Percept (%) |
|---|---|---|---|
| No Admission | No admission | 427 | 67.1 |
| | 1 Admission | 121 | 19 |
| | >/=2 Admission | 88 | 13.9 |
| Comorbid illness | Yes | 77 | 12.1 |
| | No | 559 | 87.9 |
| Type of drug | Antipsychotic | 301 | 47.3 |
| | Antidepressant | 123 | 19.3 |
| | Mood stabilizer | 16 | 2.5 |
| | Anxiolytics | 28 | 4.4 |
| | Antipsychotic & Antidepressant | 135 | 21.2 |
| | Antipsychotic and mood stabilizers | 33 | 5.2 |
| Drug side effect | Yes | 163 | 25.6 |
| | No | 473 | 74.4 |
| Counselling | Yes | 128 | 20.1 |
| | No | 508 | 79.9 |
| Duration of Rx. | < 1year | 123 | 19.3 |
| | >1yr.- <2yr. | 163 | 25.6 |
| | >2yr.- 5yr. | 151 | 23.7 |
| | >5yr. | 199 | 31.3 |
| Relapse | Yes | 412 | 64.8 |
| | No | 224 | 35.2 |
| Suicidal ideation | Yes | 250 | 39.3 |
| | No | 386 | 60.7 |
| Suicidal attempt | Yes | 126 | 19.8 |
| | No | 510 | 80.2 |
| Family Hx. MI | Yes | 144 | 22.6 |
| | No | 492 | 77.4 |
| Family Hx. Subs. | Yes | 115 | 18.1 |
| | No | 521 | 81.9 |
| Family Hx. Suicide attempt | Yes | 29 | 4.6 |
| | No | 607 | 95.4 |
| Objective severity | Mild | 493 | 77.5 |
| | Moderate | 94 | 14.8 |
| | Severe | 49 | 7.7 |
| Subjective Severity | Mild | 424 | 66.7 |
| | Moderate | 159 | 25.0 |
| | Severe | 53 | 8.3 |
| Rate your health | Excellent | 32 | 5.0 |
| | Very Good | 119 | 18.7 |
| | Good | 282 | 44.3 |
| | Fair | 156 | 24.5 |
| | Poor | 47 | 7.4 |
| **Social support variables** | | | |
| R/ship with family | Excellent | 49 | 7.7 |
| | Very Good | 147 | 23.1 |
| | Good | 299 | 47 |
| | Fair | 100 | 15.7 |
| | Poor | 41 | 6.4 |
| Family participates in Rx. | Yes | 536 | 84.3 |
| | No | 100 | 15.7 |
| Legal issues | Yes | 27 | 4.2 |
| | No | 609 | 95.8 |
| Self-esteem | Low self-esteem | 316 | 49.7 |
| | High self-esteem | 320 | 50.3 |

*(Continued)*

**Table 1.** (Continued)

| Variables | Categories | Frequency (n= 636) | Percept (%) |
|---|---|---|---|
| **Substance use variables** | | | |
| Tobacco Use | Yes | 43 | 6.8 |
| | No | 593 | 93.2 |
| Alcohol Use | Yes | 141 | 22.2 |
| | No | 495 | 77.8 |
| Khat use | Yes | 78 | 12.3 |
| | No | 558 | 87.7 |
| Cannabis Use | Yes | 04 | 0.6 |
| | No | 632 | 99.4 |
| Drug adherence | Poor adherence | 87 | 13.7 |
| | Good adherence | 549 | 86.3 |

Hx-history.

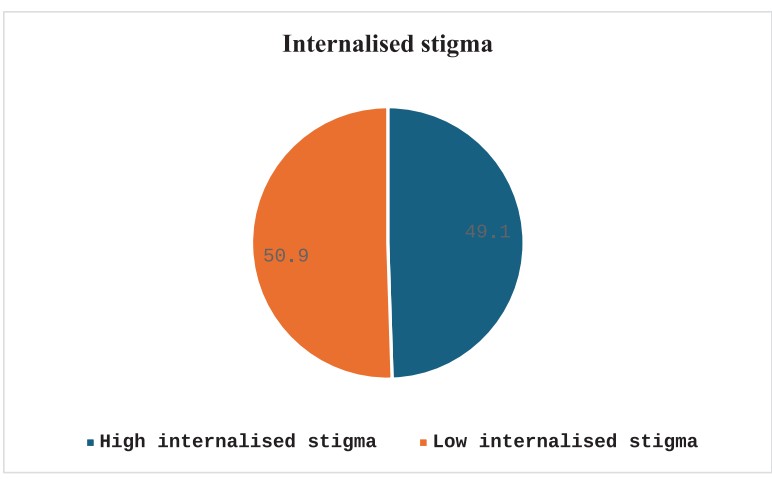

**Fig 2. Prevalence of internalised stigma in psychiatry outpatient follow-up at University of Gondar Hospital, Ethiopia, 2023(n =636).**

family history of substance use, personal perception of health fair and poor, alcohol use, having low self-esteem, and poor drug adherence.

The odds of having high internalized stigma among participants who did not have formal education was 2.19 times higher as compared to those who were educated in college/university (2.19, 95% CI:1.33,3.61). Those with suicidal ideation had almost two times higher odds of high internalized stigma when compared to patients without suicidal behavior (AOR=1.95, 95% CI:1.37,2.79). Those with a family history of substance use were 2.4 times highly likely to developing high internalised stigma than those without a family history of substance use (AOR= 2.46, 95% CI:1.54,3.93). Those reporting fair health were 3.12 times more likely to developing high internalized stigma than patients who perceived themselves as having excellent health (AOR=3.12, 95% CI:1.28,7.59), and the odds of high internalized stigma from respondents with poor personal health perceptions was nine times higher than in participants who had a perception of excellent health (AOR= 9.11, 95% CI: 2.89,28.73).

The study also found that the odds of developing high internalized stigma among patients who used alcohol was two times higher than patients who were not alcohol users (AOR= 1.89, 95%CI:1.24,2.91). The odds of developing high internalized stigma among participants

**Table 2. ISMI-9 items, people living with mental illness attending Gondar Comprehensive Specialised Hospital outpatient clinic, Ethiopia, 2023.**

| No | strongly disagree (1), disagree (2), agree (3), or strongly agree (4) | Strongly disagree (1) | Disagree (2) | Agree (3) | Strongly agree (4) |
|---|---|---|---|---|---|
| 1 | Stereotypes about the mentally ill apply to me | 21.5% | 44.7% | 28.8% | 5% |
| 2 | In general, I can live life the way I want to | 19% | 37.3% | 33.5% | 10.2% |
| 3 | Negative stereotypes about mental illness keep me isolated from the 'normal' world | 19.7% | 39.9% | 34% | 6.4% |
| 4 | I feel out of place in the world because I have a mental illness | 21.2% | 38.8% | 32.7% | 7.2% |
| 5 | Being around people who don't have a mental illness makes me feel out of place or inadequate | 18.9% | 37.7% | 34.3% | 9.1% |
| 6 | People without illness could not possibly understand me | 15.3% | 44.7% | 34.4% | 5.7% |
| 7 | Nobody would be interested in getting close to me because I have a mental illness | 17.5% | 46.7% | 28.5% | 7.4% |
| 8 | I can't contribute anything to society because I have a mental illness | 27.2% | 43.2% | 22% | 7.5% |
| 9 | I can have a good, fulfilling life, despite my mental illness | 7.2% | 14.2% | 34.6% | 44% |

with low self-esteem was 1.5 times higher compared to individuals with high self-esteem (AOR=1.55, 95% CI:1.09, 2.21).The odds of developing high internalized stigma among patients with mental illness in the outpatient clinic who had a poor drug adherent was 2.2 times higher than patients who showed good drug adherence (AOR=2.2, 95% CI:1.30,3.71) (Table 3).

## Discussion

In Sub-Saharan countries, stigma among people with mental illness is one of the most underrated consequences of common mental health problems and has received insufficient research and attention despite the risk to patients' health and impact it has on them, their families, and the entire population. However, a global review by Dubreucq et al. [53] reported that 31% of people with mental illness had high levels of internalised stigma and internalised stigma has been linked to several adverse effects, such as: worsening of psychiatric symptoms [54], decreased in seeking out mental health services and other supports, increased depression, and avoidant coping [12,14] and social avoidance [55]. In Ethiopia, the prevalence and impact of internalised stigma is not well known, is not being studied as part of mental health care and is not considered among the government priorities. In this facility-based cross-sectional study, patients with mental illness in outpatient follow-up were screened for internalised stigma and the impact of sociodemographic, clinical, social support and substance use factors based on Ethiopian context were examined for association.

This study found that the prevalence of high internalised stigma among people with mental illness at this outpatient clinics in Ethiopia was 49.1% (49.1%, 95% CI: 45, 52). Based on the current diagnosis internalised stigma for each illness 20.5% accounts for schizophrenia patients, 14.9% for depressive patients, 4.7% for bipolar patients, 2.7% for anxiety patients, 5% for other psychosis patients, 0.5% for trauma and stress-related patients, and 0.8% for somatic disorder patients. This prevalence of internalised stigma was similar to rates for schizophrenia patients in Ethiopia of 46.7% [56]. However, this prevalence was lower than another study done in Ethiopia among psychiatry patients in Addis Ababa with schizophrenia, bipolar, depression and other types diagnosis of mental illness of 61.3% [57]. Our finding was higher than previous Ethiopian studies 28% in Jimma on patient with mental illness [58], 33.5% in Addis Ababa with major depressive disorder patients [59], 24.9% for bipolar patients in Addis Ababa [41], 31.5% for mood disorder patients in Ethiopia [60], Gondar Ethiopia 27.9% [61]. Differences may relate to study site (e.g. Addis Ababa is a larger city), nature of mental illness, differences in sample size, study design, data collection method, and tools.

**Table 3. Multivariate Logistic regression on sociodemographic, clinical, substance use and social support related factors on internalised stigma among people with mental illness in Ethiopia, 2023(n =636).**

| Variables | Categories | Internalised Stigma | | COR (95% CI) | AOR (95% CI) | P-value |
|---|---|---|---|---|---|---|
| | | Low | High | | | |
| Level of education | No formal education | 55 | 84 | 2.22 (1.42,3.49)* | 2.19 (1.33,3.61)** | 0.002 |
| | Reading & writing | 24 | 22 | 1.33 (0.69,2.56) | 1.45 (0.70,2.97) | |
| | Primary school (5-8) | 51 | 56 | 1.60 (0.99,2.59) | 1.51 (0.89,2.56) | |
| | Secondary school (9-12) | 86 | 76 | 1.29 (0.84,1.97) | 1.17 (0.73,1.89) | |
| | College/ university | 108 | 74 | 1 | 1 | |
| Job of participant | Employed gov. | 51 | 38 | 1 | 1 | |
| | Private Employed | 31 | 21 | 0.9 (0.45,1.82) | 0.81 (0.34,1.91) | |
| | Farmer | 47 | 54 | 1.54 (0.86,2.73) | 1.55 (0.62,3.82) | |
| | Housewife | 45 | 59 | 1.76 (0.99,3.11) | 1.68 (0.74,3.77) | |
| | Student | 44 | 28 | 0.85 (0.45,1.60) | 1.08 (0.51,2.28) | |
| | Merchant | 41 | 40 | 1.30 (0.71,2.39) | 1.16 (0.52,2.55) | |
| | No job | 60 | 67 | 1.49 (0.86,2.58) | 1.75 (0.88,3.44) | |
| | Other | 05 | 05 | 1.34 (0.35,4.96) | 0.84 (0.18,3.82) | |
| Residence | Rural | 97 | 108 | 1.23 (0.88,1.72) | 0.82 (0.53,1.26) | |
| | Urban | 227 | 204 | 1 | 1 | |
| Age onset of illness | </= 25yrs. | 155 | 132 | 0.80 (0.58,1.09) | 1.02 (0.66,1.57) | |
| | >25yrs. | 169 | 180 | 1 | 1 | |
| Duration of illness | 6 month-5yrs. | 212 | 189 | 1 | 1 | |
| | 6yrs.-10yrs. | 72 | 69 | 1.07 (0.73,1.57) | 0.88 (0.56,1.38) | |
| | >10yrs. | 40 | 54 | 1.54 (0.96, 2.38) | 1.34 (0.78,2.27) | |
| Drug side effect | Yes | 72 | 91 | 1.41 (1.00,2.06) | 1.14 (0.74,1.75) | |
| | No | 252 | 221 | 1 | 1 | |
| Duration of Treatment | < 1year | 75 | 48 | 1 | 1 | |
| | >1yr.- <2yr. | 89 | 74 | 1.29 (0.80,2.09) | 1.25 (0.73,2.16) | |
| | >2yr.- 5yr. | 69 | 82 | 1.85 (1.14,3.01) | 1.39 (0.78,2.47) | |
| | >5yr. | 91 | 108 | 1.85 (1.17,2.92) | 1.69 (0.74,3.82) | |
| Relapse | Yes | 188 | 224 | 1.84 (1.32,2.56) | 1.34 (0.93,1.94) | |
| | No | 136 | 88 | 1 | 1 | |
| Suicidal ideation | Yes | 98 | 152 | 2.19 (1.58,3.03) * | 1.95 (1.37,2.79)** | 0.001 |
| | No | 226 | 160 | 1 | 1 | |
| Family mental illness | Yes | 59 | 85 | 1.68 (1.15,2.45) | 1.30 (0.83,2.05) | |
| | No | 265 | 227 | 1 | 1 | |
| Family substance use | Yes | 40 | 75 | 2.24 (1.47,3.42)* | 2.46 (1.54,3.93)** | 0.001 |
| | No | 284 | 237 | 1 | 1 | |
| Family suicide attempt | Yes | 11 | 18 | 1.74 (0.80,3.75) | 1.47 (0.61,3.55) | |
| | No | 313 | 294 | 1 | 1 | |
| Perception of your health | Excellent | 23 | 09 | 1 | 1 | 0.012 |
| | Very Good | 80 | 39 | 1.24 (0.52,2.94) | 1.34 (0.53,3.38) | 0.001 |
| | Good | 147 | 135 | 2.34 (1.04,5.25) | 2.11 (0.89,5.01) | |
| | Fair | 66 | 90 | 3.48 (1.51,8.02)* | 3.12 (1.28,7.59)** | |
| | Poor | 08 | 39 | 12.45 (4.21,36.79) * | 9.11 (2.89,28.73) ** | |
| R/ship with family | Excellent | 33 | 16 | 1 | 1 | |
| | Very Good | 95 | 52 | 1.12 (0.56,2.24) | 0.73 (0.33,1.57) | |
| | Good | 146 | 153 | 2.16 (1.14,4.09) | 1.12 (0.51,2.42) | |
| | Fair | 38 | 62 | 3.36 (1.63,6.91) | 1.41 (0.60,3.31) | |
| | Poor | 12 | 29 | 4.98 (2.02,12.25) | 2.03 (0.71,5.81) | |
| The family participates in the treatment | Yes | 280 | 256 | 1 | | |
| | No | 44 | 56 | 1.39 (0.90,2.13) | | |
| Legal issues | Yes | 10 | 17 | 1.80 (0.81,4.01) | 1.83 (0.75,4.45) | |
| | No | 314 | 295 | 1 | 1 | |

*(Continued)*

**Table 3.** (Continued)

| Variables | Categories | Internalised Stigma | | COR (95% CI) | AOR (95% CI) | P-value |
|---|---|---|---|---|---|---|
| | | Low | High | | | |
| Objective severity | Mild | 269 | 224 | 1 | 1 | |
| | Moderate | 38 | 56 | 1.77 (1.13,2.77) | 1.35 (0.78,2.33) | |
| | Severe | 17 | 32 | 2.26 (1.22,4.17) | 1.24 (0.5,2.59) | |
| Subjective Severity | Mild | 229 | 195 | 1 | 1 | |
| | Moderate | 77 | 82 | 1.25 (0.86,1.80) | 0.99 (0.63,1.55) | |
| | Severe | 18 | 35 | 2.28 (1.25,4.26) | 0.82 (0.36,1.87) | |
| Alcohol Use | Yes | 51 | 90 | 2.17 (1.47,3.19)* | 1.89 (1.24,2.91)** | 0.003 |
| | No | 273 | 222 | 1 | 1 | |
| Khat use | Yes | 31 | 47 | 1.67 (1.03,2.71) | 1.03 (0.55,1.90) | |
| | No | 293 | 265 | 1 | 1 | |
| Self-esteem | Low self-esteem | 193 | 127 | 2.14 (1.56,2.94)* | 1.55 (1.09,2.21)** | 0.014 |
| | High self-esteem | 131 | 185 | 1 | 1 | |
| Drug adherence | Poor adherence | 294 | 255 | 2.19 (1.36,3.51)* | 2.20 (1.30,3.71)** | 0.003 |
| | Good adherence | 30 | 57 | 1 | 1 | |

Internationally prevalence studies has also been found to vary from our findings for example, higher than global prevalence of depression self-stigma of 29% [62], 43.6% of psychiatry out patients in Singapore [63], 18.8% in Nigeria schizophrenia patient [64], 36% in USA patient with mental illness [65], but our study was lower than overseas studies like China 94.7% of schizophrenia patients [66], in Nepal 90% of patients with schizophrenia [67], 81.1% of patients in Hong Kong [68]. This difference may be accounted for country variations in cultural norms and attitude of people for mental illness are also likely important.

We identified that from sociodemographic factors examined that only level of education was associated with internalised stigma, where participants with no formal education were two times more likely to have internalised stigma than those who had been educated in college and university. Other studies in Africa have found a similar link between education and internalised stigma. For example, our systematic review and meta-analysis in Africa found that those who were unable to read and write were 3.5 times more likely than those who could report experiencing internalised stigma [69], a single study in Ethiopia also showed those who could not read and write were 3.3 times more likely than counterparts to report internalised stigma [70], in Nigeria low educational level was also found to be associated with internalised stigma [64], and in Europe, studies across 13 European countries found that education is linked to a decrease in self-stigma reports [71], In Asia research done on bipolar patients in Iran showed high degree of self-stigma associated with low level of education [72] and Turkey [17]. It may be that high levels of education may protect people from passing lowering judgments on them and likewise those who are illiterate might attribute their mental illness to supernatural causes such as demonic possession, bewitchment by an evil spirit, an ancestor's ghost, or the evil eye, which could lead to heightened internalised shame [73]. However, this is contradicted by a study done in China which found that internalised stigma was higher in educated [74]. Again, country specific factors around education systems and cultural norms likely affect internalised stigma differently.

The rapidly increasing body of research on internalised stigma has demonstrated that self-stigma is related to low self-esteem [75]. This study showed participants with low self-esteem were 1.5 times more likely to report internalised stigma than those with high self-esteem. The findings of studies carried out in several countries including Ethiopia, Israel and Taiwan have mirrored this finding [12,41,61,76–78]. Patients with severe mental illness may experience low self-esteem, which lowers their capacity to combat stigma [12]. On the other hand, even

if a person has not been personally stigmatised, the awareness that stigma exists in society can impact that person and adversely affect a person's sense of self-esteem and self-efficacy, which could change how he behave through internalising the stigma [79].

In this study, suicidal ideation was associated with internalised stigma, with those expressing suicidal ideas almost two times more likely to report internalised stigma. Similar stigmatization and social distancing processes occur to people who have suicide attempt [80]. These patients' internal stigma may also indirectly enhance their likelihood of suicidal attempt [81], the findings supportive of another study conducted in the Czech Republic on neurotic patients subjective rate of suicidality and also the objective rate of suicidality were strongly positively correlated with the internalised stigma of mental illness [82].

Another important finding was that participants with mental illness who perceived their current health as fair had three times greater odds of having high internalised stigma than those who perceived their health as being excellent. For those who perceived their health as poor this was nine times more likely. The possible reason for the association might be that the internalised stigma occurs when a patient accepts negative assumptions about mental illness and others who have it, and feels that the same assumptions may apply to them, which can then affect health [83]. In addition, being in poorer overall health may contribute to a sense of internalized stigma.

This study found that the odds of developing internalised stigma were 2.2 times higher among patients with poor drug adherence compared to participants with good drug adherence. Other studies in Addis Ababa Ethiopia [59], Czech Republic [38], the UK [84] have found a similar link. We hypothesize that this may be a result of poor follow up of patients' treatment plan, making it difficult to recover from their illness, worsening their internalized stigma [85]. The internalised stigma that limits their ability to interact with others and, in turn, reduces their follow-up visits could be another reason they avoid treatment [86].

This study found that participants who were alcohol users were almost two times more likely to report internalised stigma than those who did not use alcohol. This study is consistent with a systematic review of studies conducted in nine different countries [87], a survey conducted in Los Angeles [88], and a study about lifetime substance use conducted in Ethiopia [61].

Consistent with a study conducted in Los Angeles [88], this study also showed that having a family history of substance use was associated with a 2.46 times greater likelihood of having high internalised stigma than participants who did not have family history of substance use. Evidence exists that substance use is one of the most stigmatised behaviours [89] and alcohol and other drug users are stigmatized and socially undervalued because they are thought to be indulgent, weak-willed and lacking self-control [90,91]. Thus, stigma for people with mental illness who use alcohol and drugs may contribute to a reinforcing cycle of internalised stigma.

The finding from this study showed that the severity of mental illness had no statistical association with internalised stigma score. Our participants were patients in outpatient clinics who were stable and calm, and their condition, even when severe, may have been well managed. Similar with other findings [92,93], the current study found no association between internalised stigma and family history of mental illness. Findings on internalised stigma among people with mental illness carry significant clinical implications, particularly in Ethiopia, where structural and cultural barriers often intensify the challenges faced by this population. Addressing these issues can provide practical benefits for healthcare providers by improving patient outcomes, enhancing service delivery, and advancing the overall quality of mental health care.

## Strengths and limitations of the study

The current study used a standardized tool to measure internalised stigma, increasing the validity and reliability of the findings. Data were collected by trained and experienced psychiatry nurses and supportive supervision. However, this study has some limitations. Participants in the study

were limited to individuals using a referral hospital's outpatient programme and additionally individuals with severe cognitive impairment and impaired insight were excluded. Excluding people with severe mental illness could lead to an underestimation of the severity of stigma. While cross-sectional studies are valuable for estimating the prevalence of stigma, it is not possible to determine causality or the temporal relationship between variables and may not capture changes through time, therefore limiting the generalizability of results to other periods. Additionally, these studies are prone to confounding factors that can affect observed associations. There may be also a risk of social desirability bias because the survey was institution-based cross sectional study, and most of the data in the study were gathered through self-reported questionnaires.

## Conclusion and recommendations:

Almost one-half of the participants had a diagnosis of schizophrenia and one-third had depression. Participants faced a heavy burden of internalised stigma with about half having high internalised stigma. The high rate of internalised stigma suggests poor attitudes of Ethiopians towards mental illness. To improve the quality of life for people with mental illnesses, there needs to be significant efforts put into programs and strategies that address internalised stigma reduction among those who receive outpatient care. These strategies need to be tailored for the Ethiopian context to create awareness about internalised stigma among patients with mental health and those providing care. There was a significant association between internalised stigma scores and a number of predictor variables, reinforcing the need for particular attention to be placed on patients with no formal education, those with suicidal ideation, and those who used substances or had a family history of substance use. Patients with perception of fair and poor health, low self-esteem, and poor drug adherence would also need special considerations since these variables were found to be statistically significantly associated with high internalised stigma. We suggest for future researchers to conduct longitudinal studies to explore causal relationships between variables which had association with internalised stigma and how these factors influence stigma through time. We also suggest comparing cohorts from various time periods or regions to understand the impact of contextual factors on cause stigma as well as investigating mediating or moderating factors that may affect or increase the associations seen in cross-sectional studies.

## Ethics approval and consent to participate

Flinders University Human Research Ethics Committee approved with reference number 5416.

## Consent for publication

Not applicable.

## Acknowledgments

We want to thank all study participants, data collectors, and supervisors who contributed their time and effort to complete the research. In addition, *Wondale Getinet Alemu* thanks Flinders University and the Australian Government Research Training Programme (RTP) for funding his PhD scholarship. This research was supported by the Australian Government Research Training Program (RTP) and Flinders University.

## Author contributions

**Conceptualization:** Wondale Getinet Alemu, Lillian Mwanri, Clemence Due, Telake Azale, Anna Ziersch.

**Data curation:** Wondale Getinet Alemu.

**Formal analysis:** Wondale Getinet Alemu.

**Investigation:** Wondale Getinet Alemu.

**Methodology:** Wondale Getinet Alemu, Lillian Mwanri, Clemence Due, Telake Azale, Anna Ziersch.

**Supervision:** Wondale Getinet Alemu, Lillian Mwanri, Clemence Due, Telake Azale, Anna Ziersch.

**Visualization:** Wondale Getinet Alemu.

**Writing – original draft:** Wondale Getinet Alemu.

**Writing – review & editing:** Wondale Getinet Alemu, Lillian Mwanri, Clemence Due, Telake Azale, Anna Ziersch.

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
