## [Decision Letter · Decision Letter 0]

4 Nov 2024

PONE-D-24-17598Predictors of internalised stigma among people with mental illness attending a psychiatry outpatient clinicPLOS ONE

Dear Dr. Getinet,

Thank you for submitting your manuscript to PLOS ONE. After careful consideration, we feel that it has merit but does not fully meet PLOS ONE’s publication criteria as it currently stands. Therefore, we invite you to submit a revised version of the manuscript that addresses the points raised during the review process.

**ACADEMIC EDITOR: The manuscript contains important finding regarding the internalized stigma. However, please revise some comments raised by the reviewers giving more focus to the ethical issues, validity of the tools and editorial components.** Please submit your revised manuscript by Dec 19 2024 11:59PM. If you will need more time than this to complete your revisions, please reply to this message or contact the journal office at plosone@plos.org . Please include the following items when submitting your revised manuscript:

We look forward to receiving your revised manuscript.

Kind regards,

Yadeta Alemayehu

Academic Editor

PLOS ONE

Journal Requirements:

2. Please describe in your methods section how capacity to provide consent was determined for the participants in this study. Please also state whether your ethics committee or IRB approved this consent procedure. If you did not assess capacity to consent please briefly outline why this was not necessary in this case.

3. In the online submission form, you indicated that requesting the corresponding author

Reviewers' comments:

Reviewer's Responses to Questions

**Comments to the Author**

1. Is the manuscript technically sound, and do the data support the conclusions?

Reviewer #1: Yes

Reviewer #2: Yes

2. Has the statistical analysis been performed appropriately and rigorously? 

Reviewer #1: Yes

Reviewer #2: Yes

3. Have the authors made all data underlying the findings in their manuscript fully available?

Reviewer #1: Yes

Reviewer #2: Yes

4. Is the manuscript presented in an intelligible fashion and written in standard English?

Reviewer #1: Yes

Reviewer #2: Yes

5. Review Comments to the Author

Reviewer #1: Predictors of internalised stigma among people with mental illness attending a psychiatry outpatient clinic

Overall, the manuscript provides valuable insights into the relationship between metacognitive beliefs, mood symptoms, and fatigue in stroke survivors. It is generally well-written and organized, with a clear introduction providing context for the study. However, addressing the recommendations would enhance the clarity, robustness, and applicability of the findings.

General recommendation:

1. The study's cross-sectional design limits its ability to establish causal relationships. The manuscript acknowledges this limitation, but it would be beneficial to explicitly discuss the implications of this design choice on the interpretation of results and suggest avenues for future longitudinal research.

More specific recommendation:

Introduction

The introduction provides a thorough overview of the stigma associated with mental illness, distinguishing between internalized and perceived stigma. It effectively highlights the public health significance of stigma and the need for research in low and middle-income countries, especially Ethiopia. The literature review is comprehensive, addressing various factors contributing to stigma and noting the global relevance. However, it could benefit from a more in-depth discussion of cultural differences in stigma experiences and the rationale for choosing Ethiopia as the study site.

Methods

The methods section is well-detailed, covering the study design, sampling, and data collection procedures. The use of standardized tools, such as the ISMI-9 for measuring stigma, adds rigor to the study. However, the inclusion of more information about the training provided to data collectors and steps to minimize bias would strengthen the methodological transparency. Additionally, while the sample size calculation is justified, a discussion of potential limitations related to excluding severely impaired individuals could provide a more balanced view.

• The title of “Tools used for data collection” should be renamed to “measurement”

Results

The results are presented clearly, with appropriate use of statistical analysis to identify significant predictors of internalized stigma. The prevalence data and associations with factors like education level, suicidal ideation, and substance use are reported effectively. However, some sections could be enhanced with visual aids, such as charts or graphs, to better illustrate the relationships among variables. Additionally, discussing the clinical implications of the findings, especially for healthcare providers in Ethiopia, would add practical value.

Discussion

The discussion section successfully interprets the results in the context of existing literature, noting consistencies and differences with previous studies. It provides plausible explanations for the observed associations and acknowledges cultural factors that may influence internalized stigma. However, the discussion could be enriched by addressing the implications for policy and service development, particularly regarding mental health education and support systems. The limitations are acknowledged, but a deeper exploration of potential confounding variables would strengthen this section.

Conclusion

The conclusion effectively summarizes the key findings and reinforces the need for anti-stigma efforts in Ethiopia. It provides clear recommendations, such as early outpatient support and drug adherence interventions, but could be further strengthened by suggesting specific anti-stigma strategies tailored to the Ethiopian cultural context. Additionally, highlighting areas for future research, such as longitudinal studies to explore causality, would offer a more forward-looking perspective.

Reviewer #2: the authors did not obtain prior informed consent from the participants or their legal guardians before including them in the study. This omission violates the essential principles of ethical research involving human subjects and cannot be overlooked

6. PLOS authors have the option to publish the peer review history of their article (what does this mean? ). If published, this will include your full peer review and any attached files.

**Do you want your identity to be public for this peer review?** For information about this choice, including consent withdrawal, please see our Privacy Policy .

Reviewer #1: No

Reviewer #2: No

---

## [Author Response · Author response to Decision Letter 1]

23 Nov 2024

Response to reviewers and editors were attached as response to reviewer with other documents

---

## [Decision Letter · Decision Letter 1]

31 Jan 2025

PONE-D-24-17598R1Predictors of Internalised Stigma Among People with Mental Illness Attending a Psychiatry Outpatient Clinic in Ethiopia: Institution Based Cross Sectional study.PLOS ONE

Dear Dr. Getinet,

Thank you for submitting your manuscript to PLOS ONE. After careful consideration, we feel that it has merit but does not fully meet PLOS ONE’s publication criteria as it currently stands. Therefore, we invite you to submit a revised version of the manuscript that addresses the points raised during the review process.

**ACADEMIC EDITOR: After a careful review was undertaken by reviewers, I finally recommend you revise the document, focusing on the issues raised by the reviewers.**

We look forward to receiving your revised manuscript.

Kind regards,

Yadeta Alemayehu

Academic Editor

PLOS ONE

Journal Requirements:

Reviewers' comments:

Reviewer's Responses to Questions

**Comments to the Author**

1. If the authors have adequately addressed your comments raised in a previous round of review and you feel that this manuscript is now acceptable for publication, you may indicate that here to bypass the “Comments to the Author” section, enter your conflict of interest statement in the “Confidential to Editor” section, and submit your "Accept" recommendation.

Reviewer #1: All comments have been addressed

Reviewer #2: (No Response)

Reviewer #3: (No Response)

2. Is the manuscript technically sound, and do the data support the conclusions?

Reviewer #1: Yes

Reviewer #2: Yes

Reviewer #3: Yes

3. Has the statistical analysis been performed appropriately and rigorously? 

Reviewer #1: Yes

Reviewer #2: Yes

Reviewer #3: Yes

4. Have the authors made all data underlying the findings in their manuscript fully available?

Reviewer #1: Yes

Reviewer #2: Yes

Reviewer #3: Yes

5. Is the manuscript presented in an intelligible fashion and written in standard English?

Reviewer #1: No

Reviewer #2: (No Response)

Reviewer #3: Yes

6. Review Comments to the Author

Reviewer #1: Appreciate to the author for incorporating all the revisions based on the reviewer's comments and providing explanations for each change. The revised manuscript demonstrates improved design and writing, with a clear and cohesive introduction and discussion section. Overall, the revised manuscript meets the criteria for acceptance.

Reviewer #2: (No Response)

Reviewer #3: In the statistical analysis, it is better to add units of measurement on Age.

My additional comments are;

Regarding ethical standard as the study involves human subjects, it is better to provide ethical statement.

And on submission guidelines, the abstract should be within the limited range, it shouldn't beyond 300 words.

7. PLOS authors have the option to publish the peer review history of their article (what does this mean? ). If published, this will include your full peer review and any attached files.

**Do you want your identity to be public for this peer review?** For information about this choice, including consent withdrawal, please see our Privacy Policy .

Reviewer #1: No

Reviewer #2: No

Reviewer #3: **Yes: ** MY name is Ziyad Towfik Abdella

---

## [Author Response · Author response to Decision Letter 2]

31 Jan 2025

Thank you for your constructive comments

---

## [Editor Report · Decision Letter 2]

4 Feb 2025

Predictors of Internalised Stigma Among People with Mental Illness Attending a Psychiatry Outpatient Clinic in Ethiopia: Institution Based Cross Sectional study.

PONE-D-24-17598R2

Dear Dr. Wondale,

We’re pleased to inform you that your manuscript has been judged scientifically suitable for publication and will be formally accepted for publication once it meets all outstanding technical requirements.

Kind regards,

Yadeta Alemayehu

Academic Editor

PLOS ONE
---

## [Editor Report · Acceptance letter]

PONE-D-24-17598R2

PLOS ONE

Dear Dr. Getinet,

I'm pleased to inform you that your manuscript has been deemed suitable for publication in PLOS ONE. Congratulations! Your manuscript is now being handed over to our production team.

Kind regards,

on behalf of

Mr. Yadeta Alemayehu

Academic Editor

PLOS ONE
